# Creat3r: Confidence Reaggregation for Exploration-aware Active 3D Reconstruction

**Chih-Jung Tsai** [1 2]   **Hwann-Tzong Chen** [1 3]   **Tyng-Luh Liu** [2]

## Abstract

We present *Creat3r*, an iterative next-best-view (NBV) selection framework for efficient, high-quality 3D reconstruction. Starting from a small seed set of image–pose pairs, Creat3r repeatedly selects the most informative next camera pose. After each pose is chosen, the corresponding image is acquired and added to the multi-view set to update a 3DGS reconstruction. To guide selection, Creat3r constructs an intermediate point cloud and estimates reconstruction reliability via a novel *3D confidence field*, which is projected to candidate poses through Gaussian projection to produce *2D confidence* and *exploration maps*. These maps balance exploitation of reliable regions and exploration of uncertain or unseen areas under computational constraints. Experiments with standard 3DGS show that Creat3r consistently outperforms baselines in novel view synthesis and surface reconstruction, achieving higher SSIM and F1 scores with fewer views. The code is available at https://github.com/jimtsai23/Creat3r.

## 1. Introduction

Recent advances in 3D Gaussian Splatting (3DGS) have enabled high-quality, real-time novel view synthesis from multi-view imagery. By representing a scene as a set of differentiable 3D Gaussians, 3DGS has emerged as a leading approach for scene rendering. However, its performance still relies heavily on dense view coverage, often requiring hundreds of images per scene. This dependence creates a major practical bottleneck, increasing computational cost and optimization time, and demanding extensive manual data acquisition, especially in large-scale environments.

To reduce the acquisition burden, *active viewpoint selection* aims to identify a small yet informative subset of camera poses. Most neural rendering pipelines follow an iterative loop: optimize a scene representation from currently available views, estimate uncertainty in unobserved regions, and select the *next best view* (NBV) to reduce uncertainty. Representative methods such as FisherRF (Jiang et al., 2024) and Lyu et al. (2024) estimate uncertainty via Fisher information or variational inference.

Despite their effectiveness, existing approaches face two key limitations. First, they are tightly coupled to iterative optimization of the underlying 3D representation: each selection step typically requires re-initialization and re-optimization, leading to redundant computation. Second, many pipelines rely on global Structure-from-Motion (SfM) initialization (e.g., COLMAP) run over the entire candidate image pool. This practice introduces **information leakage** from views that are intended to remain unseen prior to selection, thereby biasing the evaluation of selection strategies. As a result, such pipelines do not fully adhere to the active selection protocol, since they implicitly assume geometric priors estimated from **all candidate views**.

To address these issues, we propose Creat3r, an efficient and robust active viewpoint selection framework that is fully decoupled from costly 3DGS optimization. Instead of repeatedly optimizing a full Gaussian scene model, Creat3r maintains a lightweight geometric proxy composed of spherical Gaussians and a sparse scaffold built directly from selected observations.

Creat3r operates through two core mechanisms. First, it incrementally estimates scene geometry by establishing pairwise correspondences between selected and candidate views, and triangulating them into 3D points via Direct Linear Transformation (DLT). This produces a dynamic scaffold that expands with each selection step, avoiding leakage from global SfM and mitigating the instability of sparse-view initialization. Second, building on this scaffold, Creat3r introduces an exploration–exploitation criterion derived from two geometry-driven maps. A *confidence map* encodes the reliability of reconstructed regions to prioritize refinement of uncertain but observed areas, while an *exploration map* highlights regions that remain unobserved or poorly constrained to encourage discovery of new content (Figure 1). Together, these signals balance local detail refinement with global scene expansion. The overall pipeline is shown in Figure 2, where geometry and confidence are re-estimated

[1]Department of Computer Science, National Tsing Hua University, Hsinchu, Taiwan [2]Institute of Information Science, Academia Sinica, Taipei, Taiwan [3]Aeolus Robotics, Hsinchu, Taiwan. Correspondence to: Tyng-Luh Liu <liutyng@iis.sinica.edu.tw>.

*Proceedings of the 43rd International Conference on Machine Learning*, Seoul, South Korea. PMLR 306, 2026. Copyright 2026 by the author(s).

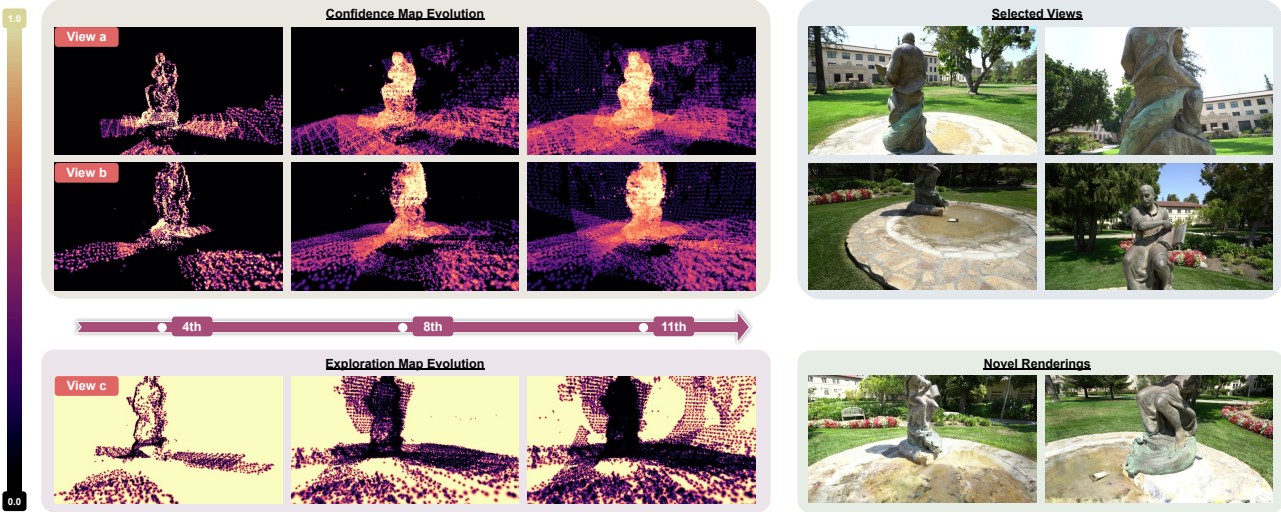

*Figure 1.* The proposed Creat3r progressively refines scene geometry and the confidence field over successive selection rounds. Left (Evolution): The top two rows (View a,b) illustrate the evolution of the *confidence maps* while the bottom row (View c) visualizes the *exploration map*. At each iteration, these maps are computed for every candidate view and jointly determine its exploration measure. The exploration map emphasizes unobserved or weakly constrained regions (bright) to encourage exploration, while the confidence map measures the reliability of reconstructed points, supporting refinement in uncertain yet already observed areas. Right (Selection and Rendering): The top four images showcase the diversity of the selected camera views, capturing various exposures and viewing angles. The bottom images display the final high-quality renderings of the reconstructed scene.

at each round to guide NBV selection.

After successive selections, Creat3r outputs a compact set of views sufficient for high-quality 3DGS reconstruction, and simultaneously produces a scene scaffold that can serve as an effective initialization for downstream tasks. Experiments demonstrate that Creat3r consistently outperforms prior viewpoint-selection methods on both novel view synthesis and surface reconstruction, achieving better quality with fewer views and lower computational cost.

Our contributions are summarized as follows:

1. We introduce Creat3r, an active viewpoint selection framework for 3DGS that is fully decoupled from iterative 3DGS optimization, yielding substantial computational savings.

2. We propose a robust geometry-based initialization that incrementally builds a sparse 3D scaffold using only selected views, avoiding the leakage bias introduced by global SfM over the full candidate pool.

3. We develop a new exploration–exploitation selection criterion based on a *confidence map* and an *exploration map* to guide NBV selection.

4. We achieve state-of-the-art performance on novel view synthesis and surface reconstruction, demonstrating improved quality and data efficiency.

## 2. Related Work

3D reconstruction with classical structure-from-motion (SfM) (Schonberger & Frahm, 2016; Pan et al., 2024) or multi-view stereo (MVS) (Schönberger et al., 2016; Yao et al., 2018) is experiencing a renaissance with the advent of emerging radiance field (Mildenhall et al., 2021; Sun et al., 2022; Fridovich-Keil et al., 2022; Müller et al., 2022), signed distance fields (Wang et al., 2021; 2023; Li et al., 2023; Liu et al., 2023), and 3D Gaussian Splatting (3DGS) (Kerbl et al., 2023; Huang et al., 2024; Dai et al., 2024).

Active 3D reconstruction determines the next best view that will most significantly enhance the quality of the reconstruction. ActiveNeRF (Pan et al., 2022) assumes parameters to be independent and estimates the uncertainty. ActiveNeuS (Ichimaru et al., 2024) extends active selection to surface reconstruction, but only for small objects. ActiveGAMER (Chen et al., 2025) uses 3DGS and RGBD inputs for next best view selection, but only in a synthetic world. NARUTO (Feng et al., 2024) and ActiveGS (Jin et al., 2025) also take RGBD inputs and extend to the real environment.

The method of Kopanas & Drettakis (2023) samples points in space and models the point camera relationship to select new views. They have adopted InstantNGP (Müller et al., 2022) to reconstruct the scene and place cameras in the empty space. FisherRF (Jiang et al., 2024) quantifies the uncertainty of each candidate using Fisher information. It uses Laplace's approximation and computes Jacobians instead

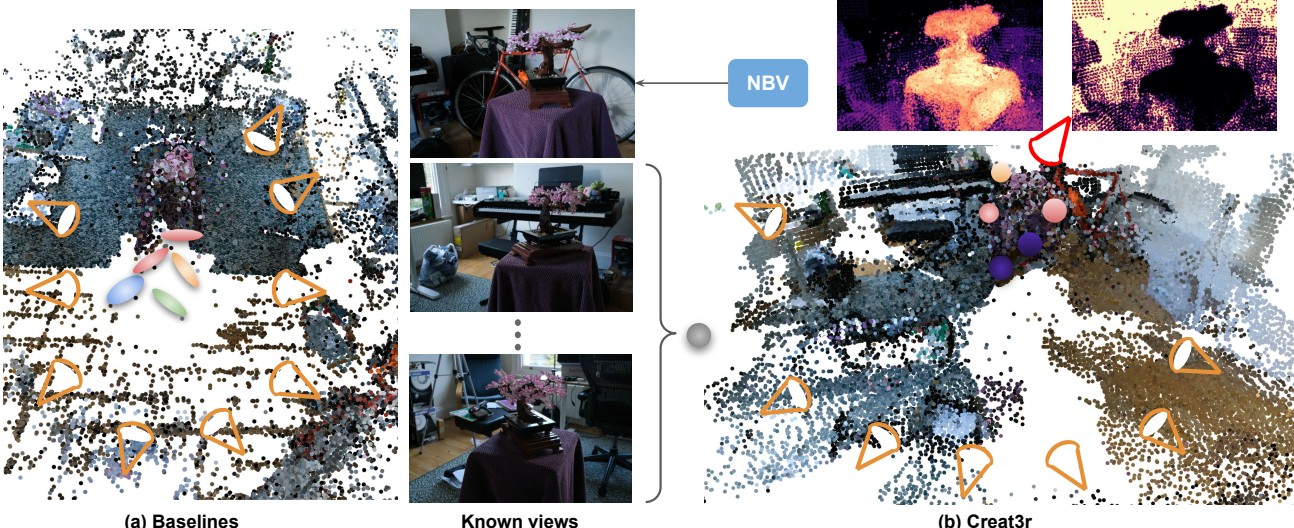

*Figure 2.* (a) Baseline methods re-initialize and re-optimize Gaussian ellipsoids at every iteration. Their initialization depends on the point cloud reconstructed with all images including the supposedly unseen views (orange cones), leading to information leakage. (b) Creat3r estimates Gaussian spheres directly from pairwise pixel correspondences. The resulting confidence field and geometry render confidence and exploration maps for each candidate viewpoint. The viewpoint with the highest exploration measure (red cone) is then selected as the *next best viewpoint* (NBV) and the newly acquired image is added to the known set.

of the Hessian matrix. It modifies the rasterize function in 3DGS to speed up computation. Similar to FisherRF, the method of Goli et al. (2024) also uses Laplace's approximation for uncertainty computation in NeRF. Like FisherRF, GauSS-MI (Xie et al., 2025) also uses information gains for view selection. Their computation does not involve known views, resulting in constant search time.

The manifold sampling technique proposed by (Lyu et al., 2024) takes a different approach and uses variational inference to model the parameter distribution of Gaussian primitives. They find an effective low-dimensional manifold that can speed up computation and a differentiable scheme to optimize uncertainty. Also relying on variance, the method of Ewen et al. (2025) computes pixel-wise higher moments for each candidate. For selection, they compute the variance for each candidate.

Unlike the methods mentioned above, we do not rely on optimization of 3DGS or NeRF during view selection. Instead, we use 2D correspondence predictions (Lindenberger et al., 2023; Sarlin et al., 2020; Sun et al., 2021; Leroy et al., 2024) to estimate robust geometry for active view selection.

## 3. Method

**Problem Setting.** We consider an *active viewpoint selection* protocol in which a pool of candidate camera poses is known *a priori*, but the corresponding images are *not* accessible until a pose is selected. At round $t$, the algorithm chooses a next pose $W_t$ from the remaining candidate set,

acquires the associated image $I_t$, and adds the newly revealed pair $(I_t, W_t)$ to the set of observed views. A 3DGS reconstruction is then updated using only the images that have been acquired so far. The goal is to reach high-quality reconstruction with as few selected views as possible.

**Pitfall: Global SfM Leakage.** Given only a candidate pose pool, an active method must predict which pose is most informative *without* using image content from unselected viewpoints. The benchmark of Lyu et al. (2024) simulates this constraint on Mip-NeRF 360 by revealing pixels only after selection. However, their pipeline initializes geometry using a global Structure-from-Motion (SfM) point cloud computed from *all* images in the candidate pool. This introduces information leakage: geometric priors derived from unseen views are implicitly available to the selection policy, violating the active vision protocol and biasing the evaluation. We therefore refine the benchmark by replacing global SfM initialization with leakage-free initialization schemes that depend only on the currently selected views.

**Formulation.** Let $\mathcal{S} = \{(I_i, W_i)\}_{i=1}^N$ denote the set of image–pose pairs for a scene, where selecting a viewpoint means choosing a pose $W_i$ and then acquiring its image $I_i$. We start from a small *known* set $\mathcal{S}^K = \{(I_i, W_i)\}_{i \in \mathcal{K}}$ and an *unseen candidate* set $\mathcal{S}^C = \{(I_j, W_j)\}_{j \in \mathcal{C}}$, such that $\mathcal{S} = \mathcal{S}^K \dot{\cup} \mathcal{S}^C$ and $\mathcal{K} \cap \mathcal{C} = \emptyset$. At each round, the algorithm selects a pose from $\{W_j\}_{j \in \mathcal{C}}$ and only then observes the corresponding image, which is used to expand $\mathcal{S}^K$. The procedure repeats until the view budget is reached, while

the total number of 3DGS optimization iterations is limited and distributed across rounds.

### 3.1. Robust Point Estimation

During selection, we maintain an intermediate 3D estimate $\tilde{\mathcal{P}}$ computed from the currently observed views $\mathcal{S}^K$. This estimate serves two roles: it provides a proxy for expected reconstruction quality, and it underpins the viewpoint scoring criterion described in later sections.

Given $\mathcal{S}^K$, we extract pixel correspondences for all image pairs in $\mathcal{S}^K$ using a learned matcher (e.g., LightGlue (Lindenberger et al., 2023)) and triangulate them to recover 3D points via stereo geometry. Consider two observed images $I_a$ and $I_b$ captured by cameras $a$ and $b$, respectively. We denote the resulting set of corresponding pixels as

$$\{(u_a, v_a) \leftrightarrow (u_b, v_b)\}. \tag{1}$$

For camera $a$ with projection matrix $\mathbf{P}^a$, the 3D-to-2D projective relation is

$$\kappa_a \begin{bmatrix} u_a \\ v_a \\ 1 \end{bmatrix} = \mathbf{P}^a \begin{bmatrix} x \\ y \\ z \\ 1 \end{bmatrix}, \tag{2}$$

where $\kappa_a$ is a depth scale factor. The same equation holds for camera $b$ by replacing $\mathbf{P}^a$ with $\mathbf{P}^b$ and $(u_a, v_a)$ with $(u_b, v_b)$.

To eliminate $\kappa$, we apply the standard Direct Linear Transform (DLT) construction. For each view, we multiply the third row by $u$ (or $v$) and subtract it from the first (or second) row, producing a linear system of the form

$$\mathbf{A}[x, y, z]^T = \mathbf{b}, \tag{3}$$

with

$$\mathbf{A} = \begin{bmatrix} u_a\mathbf{P}^a_{31} - \mathbf{P}^a_{11} & u_a\mathbf{P}^a_{32} - \mathbf{P}^a_{12} & u_a\mathbf{P}^a_{33} - \mathbf{P}^a_{13} \\ v_a\mathbf{P}^a_{31} - \mathbf{P}^a_{11} & v_a\mathbf{P}^a_{32} - \mathbf{P}^a_{12} & v_a\mathbf{P}^a_{33} - \mathbf{P}^a_{13} \\ u_b\mathbf{P}^b_{31} - \mathbf{P}^b_{11} & u_b\mathbf{P}^b_{32} - \mathbf{P}^b_{12} & u_b\mathbf{P}^b_{33} - \mathbf{P}^b_{13} \\ v_b\mathbf{P}^b_{31} - \mathbf{P}^b_{11} & v_b\mathbf{P}^b_{32} - \mathbf{P}^b_{12} & v_b\mathbf{P}^b_{33} - \mathbf{P}^b_{13} \end{bmatrix}, \tag{4}$$

and

$$\mathbf{b} = \begin{bmatrix} \mathbf{P}^a_{14} - u_a\mathbf{P}^a_{34} \\ \mathbf{P}^a_{24} - v_a\mathbf{P}^a_{34} \\ \mathbf{P}^b_{14} - u_b\mathbf{P}^b_{34} \\ \mathbf{P}^b_{24} - v_b\mathbf{P}^b_{34} \end{bmatrix}. \tag{5}$$

We solve for the 3D point via a DLT-based least-squares estimate, $[x, y, z]^T = (\mathbf{A}^T\mathbf{A})^{-1}\mathbf{A}^T\mathbf{b}$ and apply this triangulation to all correspondences across every image pair in $\mathcal{S}^K$. This produces a set of *co-visible* 3D points whose coordinates are constrained by stereo geometry, providing a stable scaffold for active selection.

### 3.2. Confidence Field via Reaggregation

Given the intermediate 3D reconstruction $\tilde{\mathcal{P}}$, we assign each point a confidence score that measures its reliability under the currently observed views, and use these scores to guide the next-best-view selection among candidate poses.

Let a point $p \in \tilde{\mathcal{P}}$ have position $\mathbf{p}$ and an associated color $\mathbf{c}$ (e.g., the average of its observed colors). For each observed camera $a \in \mathcal{S}^K$, we define a binary visibility indicator

$$g(a, p) \in \{0, 1\}, \quad g(a, p) = 1 \text{ iff } p \text{ is visible in camera } a. \tag{6}$$

Visibility alone is insufficient, since occlusions, clutter, and mismatched correspondences can yield unreliable points. We therefore measure *photometric consistency* by projecting $p$ into each supporting camera and comparing the projected pixel color to $\mathbf{c}$. Specifically, we define

$$H(p) = \exp\left(-\frac{1}{\sum_{a \in \mathcal{S}^K} g(a, p)} \sum_{a \in \mathcal{S}^K} g(a, p) \|\mathbf{c}_a(p) - \mathbf{c}\|_2\right), \tag{7}$$

where $\mathbf{c}_a(p)$ is the pixel color at the projection of $p$ in camera $a$. If $\sum_{a \in \mathcal{S}^K} g(a, p) = 0$, we set $H(p) = 0$.

We then define the **3D confidence field** $\mathcal{M}_{\text{Conf}}$ by combining photometric consistency with multi-view support:

$$\mathcal{M}_{\text{Conf}}(p) = H(p) \cdot \bar{g}(p), \quad \bar{g}(p) = \frac{1}{|\mathcal{S}^K|} \sum_{a \in \mathcal{S}^K} g(a, p). \tag{8}$$

After each selection round, we rebuild $\tilde{\mathcal{P}}$ from the updated $\mathcal{S}^K$ and reaggregate visibility and photometric consistency to update $\mathcal{M}_{\text{Conf}}$, ensuring that the confidence field reflects evidence from the currently selected views in $\mathcal{S}^K$.

### 3.3. View-Specific Confidence and Exploration Maps

Thus far, we have constructed a geometry scaffold $\tilde{\mathcal{P}}$ and its confidence field $\mathcal{M}_{\text{Conf}}$ from the observed set $\mathcal{S}^K$. We next propagate these 3D quantities to each candidate pose $W \in \{W_j\}_{j \in \mathcal{C}}$ to obtain two view-specific 2D maps: a *confidence map* and an *exploration map*. These maps quantify how much a candidate view is expected to refine already observed regions versus reveal new content.

We adopt a simplified 3DGS-style projection. Each point $p \in \tilde{\mathcal{P}}$ is represented as an isotropic Gaussian centered at $\mathbf{p}$ with radius $r$:

$$G(\mathbf{x}) = o \cdot \exp\left(-\frac{1}{2r^2} \|\mathbf{x} - \mathbf{p}\|^2\right), \tag{9}$$

where $o$ is a fixed opacity. In the isotropic case, the projected radius scales as $r^{2D} = r \cdot f/\lambda$, where $f$ is the focal length and $\lambda$ is the depth of the point relative to the *rendering view*

(i.e., the camera used to produce the point $\mathbf{p}$). We set $r$ such that the projected footprint in the rendering view is approximately one pixel, and we alpha-composite depth-sorted Gaussians to obtain the per-view maps.

From the geometry scaffold $\hat{\mathcal{P}}$ and its confidence field $\mathcal{M}_{\mathrm{Conf}}$, we render two view-specific maps for each candidate pose $W$. We first compute a **2D confidence map** $\mathbf{M}_{\mathrm{Conf}}(W)$ by projecting $\hat{\mathcal{P}}$ into view $W$ with the simplified Gaussian renderer, assigning each point $p \in \hat{\mathcal{P}}$ an intensity equal to its confidence $\mathcal{M}_{\mathrm{Conf}}(p) \in [0, 1]$. We then compute a **2D exploration map** $\mathbf{M}_{\mathrm{Exp}}(W)$ by rendering the same point set with unit intensity and inverting the resulting image, so that pixels with little or no projected support (i.e., unobserved or weakly constrained regions) receive higher values. Unless otherwise specified, we use a fixed opacity $o = 0.8$ in Eq. (9) for all renderings.

### 3.4. Exploration Measure

The two maps provide a quantitative score for each candidate viewpoint. Intuitively, a good next view should cover regions that remain unexplored, while avoiding views that primarily revisit already confident content. We formalize this with

$$\text{Exploration}(W) = \Phi_{\mathrm{Exp}}(W) \;-\; \tau \cdot \overline{\mathbf{M}}_{\mathrm{Conf}}(W) \,, \quad (10)$$

where $\Phi_{\mathrm{Exp}}(W)$ represents the amount of unexplored content visible from $W$, formulated as

$$\Phi_{\mathrm{Exp}}(W) = \frac{1}{|\mathcal{D}|} \sum_{q \in \mathcal{D}} \mathbb{1}\left(\mathbf{M}_{\mathrm{Exp}}(q; W) > 0\right) \,. \quad (11)$$

Here, $\mathbb{1}(\cdot)$ denotes the indicator function and $\mathcal{D}$ is the pixel set of $\mathbf{M}_{\mathrm{Exp}}$. $\overline{\mathbf{M}}_{\mathrm{Conf}}(W)$ penalizes views that mostly observe already confident (well-covered) regions, which computes the conditional spatial mean over the valid non-zero regions $\Omega^+$:

$$\overline{\mathbf{M}}_{\mathrm{Conf}}(W) = \frac{1}{|\Omega^+|} \sum_{(u,v) \in \Omega^+} \mathbf{M}_{\mathrm{Conf}}^{(u,v)}(W) \,, \quad (12)$$

where $\Omega^+ = \{(u,v) \mid \mathbf{M}_{\mathrm{Conf}}^{(u,v)}(W) > 0\}$. The balancing parameter $\tau$ controls the trade-off between exploration and refinement. We select the next viewpoint by

$$W^* = \underset{W \in \{W_j\}_{j \in \mathcal{C}}}{\arg\max} \; \text{Exploration}(W). \quad (13)$$

After choosing $W^*$, we acquire its image $I^*$, add $(I^*, W^*)$ to $\mathcal{S}^K$, and repeat. Importantly, the selection step depends only on $\mathcal{S}^K$ and the candidate poses $\{W_j\}_{j \in \mathcal{C}}$; the image content of $I^*$ remains unseen until *after* $W^*$ is decided.

## 4. Experiment

To evaluate our method, we perform comprehensive comparisons on 3D reconstruction. The first task is novel view

synthesis. The experimental setting follows previous methods, and the results are detailed in Section 4.3. The second task is surface reconstruction. This is a more severe task and has not been discussed by previous active view-selection methods. It is shown that our method is capable of reconstructing the surface under limited views. Further discussion is presented in Section 4.4.

### 4.1. Implementation Details

Creat3r uses pixel correspondences to estimate robust geometry. Any correspondence estimation method can be used in our framework. In the experiment, we report the evaluation with two different correspondence estimation methods, LightGlue and MASt3R, for novel view synthesis. Note that we simply treat MASt3R as a correspondence network for pixel matching. We do not use their point estimate in the entire process of view selection.

Our framework is 3D model agnostic, meaning it can accept any 3D reconstruction technique. For a fair comparison, we use 3DGS as our 3D representation method, following the baselines. Since Creat3r provides a reliable 3D scaffold, 3DGS converges in a very short time: We finish 3DGS optimization in 5,000 iterations. We also discuss computational efficiency comparison in the appendix. As shown in Table A.1, Creat3r achieves significantly faster selection times.

### 4.2. Experimental Details

We compare Creat3r with the previous state of the arts. Specifically, we consider FisherRF (Jiang et al., 2024), Lyu et al. (2024) and Kopanas & Drettakis (2023) as competitive counterparts. Since the method of Kopanas & Drettakis (2023) is originally designed for NeRF, we adapt their method to 3DGS. We also use the 3DGS version of FisherRF in a single selection manner.

All of the baseline methods have unified searching and optimization iterations. We follow the setting of Lyu et al., which takes 20,000 iterations for searching. They take another 10,000 iterations for final optimization. In addition to the uncertainty estimation approaches, we construct a simple baseline through farthest point sampling (FPS). This method only considers the position of each camera and collects the views with the largest inner distances. Despite the absence of camera orientation, this strategy can be useful, especially in an inward-captured dataset.

For novel view synthesis, we use Mip-NeRF 360 dataset as the benchmark. For every scene, $1/8$ of the views are split as novel views. The remaining views form the candidate set. Each scene has three initial views as the known set. We collect the known set as described in ReconFusion (Wu et al., 2024). Like Lyu et al., we set the active selection to

*Table 1.* Novel view synthesis evaluation on Mip-NeRF 360 dataset. We present the evaluations of 3DGS optimized with 10 and 20 selected views. (*) denotes methods initialized with COLMAP-induced subsampled points. (‡) indicates methods initialized with Creat3r-LightGlue points, and (†) indicates methods initialized with Creat3r-MASt3R. Best and second-best results are highlighted.

| Method | 10 cameras | | | 20 cameras | | |
|---|---|---|---|---|---|---|
| | PSNR↑ | SSIM↑ | LPIPS↓ | PSNR↑ | SSIM↑ | LPIPS↓ |
| FPS* | 12.529 | 0.261 | 0.613 | 14.918 | 0.389 | 0.528 |
| FPS† | 13.560 | 0.362 | 0.555 | 14.940 | 0.439 | 0.514 |
| FisherRF (Jiang et al., 2024)* | 12.625 | 0.264 | 0.608 | 15.434 | 0.390 | 0.515 |
| FisherRF (Jiang et al., 2024)† | 14.196 | 0.392 | 0.546 | 16.028 | 0.474 | 0.491 |
| Kopanas & Drettakis (2023)* | 13.022 | 0.284 | 0.596 | 15.658 | 0.407 | 0.506 |
| Kopanas & Drettakis (2023)† | 13.677 | 0.383 | 0.546 | 15.727 | 0.470 | 0.499 |
| Lyu et al. (2024)* | 12.561 | 0.264 | 0.612 | 15.446 | 0.401 | 0.518 |
| Lyu et al. (2024)† | 14.282 | 0.377 | 0.547 | 16.264 | 0.503 | 0.480 |
| Creat3r ‡ | 16.040 | 0.449 | 0.536 | 19.637 | 0.567 | 0.443 |
| Creat3r † | 17.809 | 0.511 | 0.523 | 20.678 | 0.601 | 0.397 |

gather 10 and 20 views for 3DGS optimization.

For surface reconstruction, we use the Tanks&Temples dataset as the benchmark. We follow the practice of GOF (Yu et al., 2024) and sample three scenes from the dataset's training set for evaluation. For each scene, $1/8$ of the views are split as novel views. We assign three views in each scene as the initial known set. The number of view collections is set to 20.

### 4.3. Novel View Synthesis

The evaluation process for novel view synthesis includes several steps. First, each method selects a certain number of views for 3DGS optimization. The optimized model renders images in novel poses. The evaluation compares the image quality between rendered and ground-truth images. Standard metrics include peak signal-to-noise ratio (PSNR), structural similarity index (SSIM), and learned perceptual image patch similarity (LPIPS). The three metrics reflect different perspectives of image quality. When an image has better quality, it should have higher PSNR and SSIM, and also lower LPIPS. In this task, we use the popular MipNeRF-360 dataset for evaluation. The dataset contains nine real-world scenes, including indoor and outdoor captures. To fully expose the data efficiency of each method, the number of selections is set to 10 and 20, respectively. In the original data split, each scene has hundreds of views for optimization. Here, the training set is used as the candidate pool to find the optimal selections.

In previous literature, the evaluation process includes the sparse reconstruction from SfM. As mentioned earlier, the point cloud is reconstructed from hundreds of views that are actually treated as candidates during the selection. Using this point cloud for 3DGS initialization would reveal geometric information and lead to unfair/biased comparisons.

To develop a fair comparison, we provide three different kinds of initialization. The first one is subsampling initialization. To prevent unlimited space sampling, we use the extreme values of the SfM point cloud coordinates (induced by COLMAP) as boundaries and sample within them. We are also interested in the case where other competing methods have the same initial points as ours. The second and third initializations share the same point sets estimated with Creat3r and initial views. Each of them relates to LightGlue and MASt3R-matching, respectively.

The evaluation results are listed in Table 1. A more complete comparison can be found in the Table A.2. We use (*) to indicate that the methods use COLMAP-induced sampling initialization. (‡) and (†) indicate the methods start with robust points generated with Creat3r and initial views. Each baseline method reports two results with COLMAP-induced subsampling and Creat3r-MASt3R-matching initialization. Due to space limit, we report the baseline results with Creat3r-LightGlue in the appendix. Our Creat3r-MASt3R-matching initialization has consistent improvements in all the baselines, compared to the COLMAP-induced subsampling counterpart.

For different active selection strategies, FPS shows basic performance as it only considers the positions of the camera and ignores the orientations. FisherRF and Lyu et al. have similar performance. FisherRF performs better when the view is sparse. The method finds a candidate with the most information gain and achieves a significant improvement in the initial selections. The manifold sampling method of Lyu et al. performs better when there are more views. They use posterior to the scene, which is more accurate when there are more observations.

The method of Kopanas & Drettakis considers the visibility and viewing directions of sampled points. Their method has

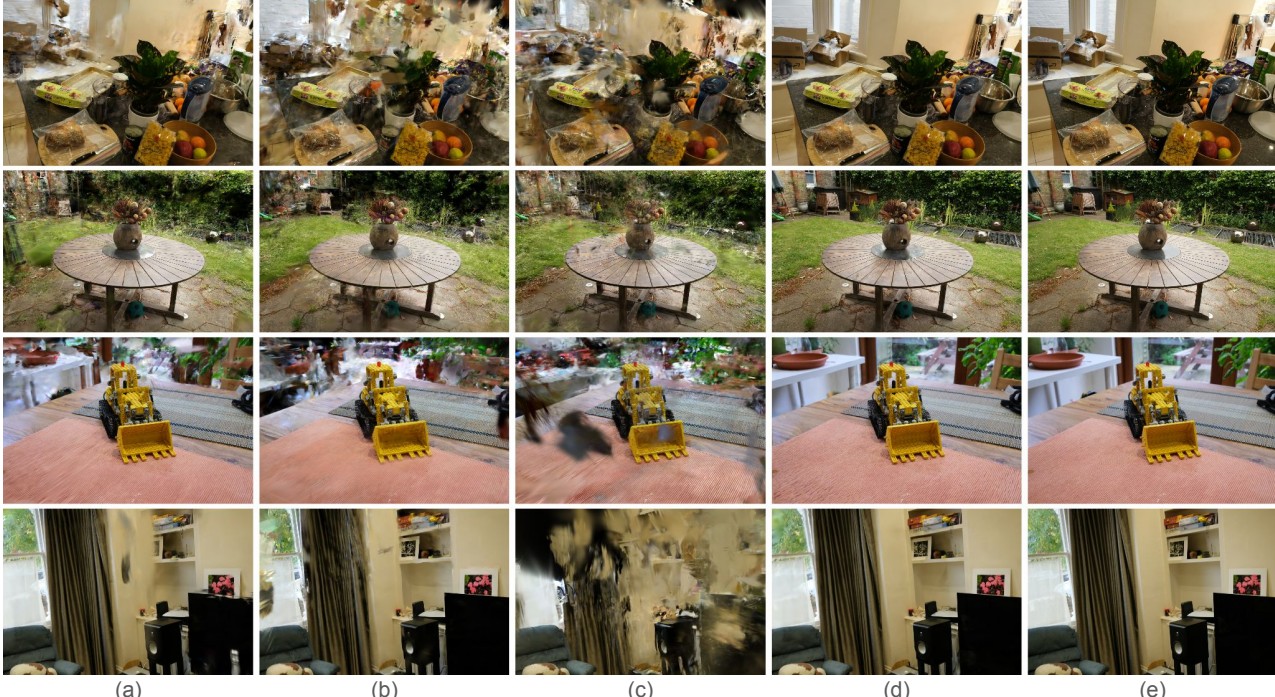

*Figure 3.* Qualitative comparison of active view selection on the Mip-NeRF 360 dataset for 20 selected views. The results demonstrate novel view renderings produced by competing methods: (a) FisherRF (Jiang et al., 2024), (b) Kopanas & Drettakis (2023), (c) Lyu et al. (2024), and (d) our proposed method, Creat3r. Column (e) serves as the ground truth novel view reference. The comparison highlights the superior ability of Creat3r to synthesize high-fidelity views with finer details and fewer artifacts.

better performance in indoor or area-constrained scenes. Interestingly, the method of Kopanas & Drettakis outperforms FisherRF and Lyu et al. when the initialization is COLMAP-induced subsampling, while FisherRF and Lyu et al. surpass Kopanas & Drettakis when using Creat3r estimated points as initialization. This suggests that uncertainty estimation, either with information gain or variational inference, is more beneficial from robust geometry, while Kopanas & Drettakis is less dependent on geometry.

Creat3r outperforms all baselines in all metrics, regardless of the number of selections. Due to its robust geometry, our 3D representation requires only half of the iterations for optimization. Note that optimization is difficult due to sparse views and the absence of ground-truth points. Creat3r estimates robust geometry, projects confidence and exploration maps to each candidate, and carefully selects the next best view by exploration measure. All efforts significantly improve the novel view quality. The comparison validates that our design is effective in various real-world scenes.

Creat3r can be seamlessly paired with any correspondence network. Creat3r-MASt3R consistently outperforms Creat3r-LightGlue in all metrics. The key difference lies in the scaffold statistics. In the 10-view Mip-NeRF 360 evaluation, MASt3R provides about 67k scaffold points on average by round 10, while LightGlue provides about 19k.

This difference directly affects the quality of the scaffold, confidence field, and subsequent view selection. More generally, sparse matchers such as SuperGlue or LightGlue yield fewer correspondences, whereas denser approaches such as LoFTR, RoMa, or MASt3R can recover many more matches, including in weakly textured regions such as walls or ceilings. Leveraging dense matching frameworks significantly enhances the reconstruction fidelity of Creat3r.

The qualitative results are illustrated in Figure 3. The figure demonstrates novel view renderings of four independent scenes in Mip-NeRF 360 dataset. The comparison highlights the superior ability of Creat3r to synthesize high-fidelity views with finer details and fewer artifacts. Other competing methods render with some artifacts due to a suboptimal selection set. More qualitative comparisons and the selected sequence comparisons are demonstrated in Section A.3 and Section A.4 of the appendix.

### 4.4. Surface Reconstruction

The task aims to reconstruct the actual surface of the scene. While 3DGS does not produce an actual surface, we adopt the mesh extraction pipeline from 2DGS (Huang et al., 2024). After optimization, we render depth maps for selected views. The depths are then fused into a voxel grid using truncated signed distance fusion (Curless & Levoy,

*Table 2.* Surface reconstruction evaluation on Tansks&Temples dataset. **Best** results are highlighted. The approach of Kopanas & Drettakis (2023) suffers from extremely limited surface coverage (low recall), resulting in a low F1-score. In contrast, Creat3r maintains a superior balance between precision and recall.

|  | Prec.(%) | Rec.(%) | F1(%) |
|---|---|---|---|
| FisherRF (Jiang et al.) | 7.91 | 9.81 | 8.61 |
| Kopanas & Drettakis | **17.6** | 0.58 | 0.84 |
| Lyu et al. | 5.85 | 5.86 | 5.61 |
| Creat3r | 14.09 | **25.93** | **18.05** |

1996) and extracted via marching cubes (Lorensen & Cline, 1987). The evaluation process densely samples the reconstructed surface and compares it against the ground truth. The predicted point is considered valid if it is within a 5-millimeter distance from the ground-truth points. The reported metrics are precision, recall, and F1-score.

In this evaluation, we use the popular Tanks&Temples dataset as the benchmark. Our setting is similar to GOF (Yu et al., 2024), which samples three scenes for evaluation, including `Caterpillar`, `Ignatius`, and `Truck`. The scenes are more difficult than Mip-NeRF 360 scenes and exhibit a wide variety of lighting conditions, such as sunshine and reflective surfaces. Each scene provides the surface ground truth of the foreground object. Only the foreground surface is evaluated. To reconstruct the surface, all the competitors must find optimal view collections of 20 views that cover most of the appearance and regional detail.

The results are shown in Table 2. Note that the numerical values are shown in percentages. Since the three scenes are outdoors and have different exposures across views, it is challenging for 3DGS to model the scene, as the optimization solely relies on appearance differences. While the methods of FisherRF, Kopanas & Drettakis, and Lyu et al. depend on optimized 3DGS for view selection, they face challenges when the optimization fails. On the other hand, Creat3r is not affected by 3DGS performance. Although our method yields lower precision compared to Kopanas & Drettakis, their approach suffers from extremely limited surface coverage (low recall), resulting in a compromised F1-score. In contrast, Creat3r maintains a superior balance between precision and recall.

The robust geometry enables view selection even in challenging scenes, especially for luminance variation across views. To further validate the point, we compare Creat3r with the original 3DGS. To prevent the influence of sparse views, the 3DGS optimization uses the entire training set, which includes 200 to 400 images. The comparison is listed in Table 3. The result aligns with our point. The scene `Caterpillar` is captured in the rural field. The images have severe exposure differences. Our method is not af-

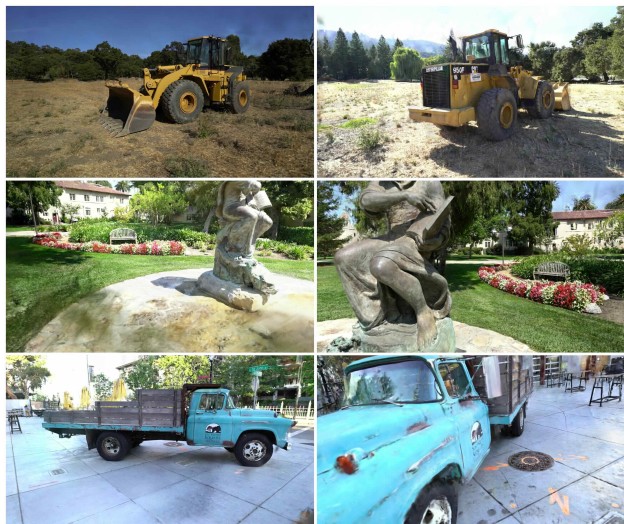

*Figure 4.* Novel view rendering of Tanks&Temples scenes. Even though the lighting condition varies drastically in different views, Creat3r can still reconstruct the surface of the targets with only 20 views.

*Table 3.* F1-score comparison between Creat3r and original 3DGS (Kerbl et al., 2023). Creat3r only optimizes with 20 selected views. 3DGS uses all of the training set for optimization (more than 200.)

|  | Caterpillar | Ignatius | Truck |
|---|---|---|---|
| 3DGS | 0.08 | 0.04 | **0.19** |
| Creat3r | **0.10** | **0.25** | **0.19** |

fected by this adversary. On the other hand, `Ignatius` has a sculpture in the foreground. The material of the sculpture reflects specular light, which leads to inconsistencies across views. Creat3r has the same performance with 3DGS on the `Truck` scene, while 3DGS uses 11 times more images for training. It suggests that our selection criterion is effective and provides data efficiency for 3DGS optimization. Novel view renderings are shown in Figure 4.

### 4.5. Ablation Study

In the ablation study, we focus on three components of Creat3r: namely, robust point estimate, confidence reaggregation, and exploration. The study excludes one component at a time, evaluating the performance drop for the exclusion. The evaluation uses nine scenes in the Mip-NeRF 360, and the number of selections is set to 10.

The comparison is shown in Table 4. The first row excludes the point estimate (Pt.). Instead, we use MASt3R predicted points as an alternative. Empirically, the primary bottleneck for the MASt3R-predicted scaffold is not just point noise, but global geometric bias. Specifically, MASt3R

*Table 4.* Ablation study of Creat3r. Influence comparison of robust point estimate (Pt.), exploration (Expl.), and confidence reaggregation (Conf.). We use nine different scenes in Mip-NeRF 360 for evaluation. The first row excludes the point estimate and, instead, uses MASt3R predicted points as an alternative.

| Pt. | Expl. | Conf. | PSNR↑ | SSIM↑ | LPIPS↓ |
|-----|-------|-------|-------|-------|--------|
|     | ✓     | ✓     | 16.479 | 0.458 | 0.574 |
| ✓   |       | ✓     | 17.022 | 0.507 | 0.525 |
| ✓   | ✓     |       | 17.457 | 0.502 | 0.543 |
| ✓   | ✓     | ✓     | **17.809** | **0.511** | **0.523** |

often introduces relative depth shifts between foreground and background because it does not explicitly utilize known poses during inference. Compared to our full model in the fourth row, the robust point estimate is the most effective technique, yielding an improvement of 1.26 in PSNR.

Exploration (Expl.) and confidence reaggregation (Conf.) provide different aspects of improvement. While confidence reaggregation performs better in SSIM and LPIPS metrics, the exploration has a higher performance in PSNR. In the experiment, we find that exploration performs better in constrained scenes, such as indoor environments. On the other hand, confidence performs better on outdoor scenes. With this in mind, we achieve the best of both worlds through the exploration measure and obtain an overall better performance. The study also reflects different choices of the balancing parameter $\tau$. When $\tau = +\infty$, the exploration term is removed (second row). When $\tau = 0$, the confidence term is excluded (third row). The full model uses $\tau = 1$ (fourth row, highlighted), which best synergizes exploration and confidence. We also note that a linear combination is a natural and transparent design here. Empirically, the exploration term dominates in the early rounds because many regions remain unobserved, whereas the confidence term becomes more influential in later rounds as the scaffold grows and refinement becomes more important. We present a qualitative evaluation of the ablated components in Section A.5. Figure 7 illustrates the full Creat3r model alongside its ablated variants, highlighting their respective functionalities. Other hyperparameter choices are discussed in Section A.6.

### 4.6. Limitations and Discussion

A primary limitation of Creat3r stems from its reliance on geometric co-visibility between the candidate views and the current reconstruction scaffold. While our approach proves highly effective for inward-facing (object-centric) and forward-facing trajectories, outward-facing scenarios (e.g., the `room` scene) present a distinct challenge. In such cases, candidate views often observe disjoint regions of the scene and may share minimal overlap with the initial estimated geometry. Consequently, these views yield null confidence maps and uniformly high-intensity exploration

maps. This ambiguity can inadvertently bias the exploration measure towards redundant sampling of unconstrained regions, leading to suboptimal convergence. We emphasize that this vulnerability is inherent to the active selection paradigm; all baseline methods similarly struggle to identify informative views in the absence of initial geometric overlap. To mitigate this, we implement a regularization strategy that temporarily masks such candidates from the selection pool. These views are subsequently reintroduced as the confidence field expands and sufficient geometric connectivity is established to meaningfully constrain them.

Since Creat3r does not rely on iterative 3DGS optimization for view selection, it inherently generalizes to alternative reconstruction paradigms, such as feed-forward frameworks or other optimization-based alternatives. In Section A.7, we demonstrate this versatility by integrating Creat3r into a feed-forward pipeline, which enhances the reconstruction quality of Depth Anything 3. We also study the influence of Creat3r on surface reconstruction with geometry-oriented representations, such as 2DGS. Creat3r-2DGS outperforms Creat3r-3DGS in sparse view settings, as well as 2DGS with complete input views. Meanwhile, our framework can be extended naturally to online viewpoint generation. Creat3r maintains an intermediate geometry scaffold and a 3D confidence field, and evaluates viewpoints through projected confidence and exploration maps. This same idea could be applied beyond a fixed pool by scoring poses sampled from a continuous camera space. A practical strategy is a two-stage scheme: first, generate coarse candidate poses from a simulator, motion planner, or local camera search around under-observed regions; then, refine the best pose by locally perturbing position and orientation and re-evaluating the same exploration score. Ultimately, the adaptability demonstrates the immense potential of Creat3r as an active perception engine for embodied intelligence.

### 5. Conclusion

The proposed Creat3r is a novel active view selection framework for 3D reconstruction. Unlike prior methods that rely on iterative optimization of the 3D representation, Creat3r is fully decoupled from this process, leading to substantial computational savings. Our method builds a robust 3D scaffold from only the selected views, avoiding the information leakage bias inherent in global SfM pipelines. Through the use of confidence map and exploration map, Creat3r balances local detail refinement with global scene expansion, achieving state-of-the-art results for novel view synthesis and surface reconstruction. The reliable geometry obtained by Creat3r also reduces the optimization time for downstream tasks, such as 3DGS, providing a significant step toward making high-quality 3D reconstruction feasible with a minimal and informative set of images.

## Acknowledgements

This work was supported in part by NSTC Graduate Research Fellowship, NSTC grants 112-2221-E-A49-100-MY3, 113-2221-E-001-010-MY3, 115-2634-F-007-003 of Taiwan. We thank the National Center for High-performance Computing for providing computing resources.

## Impact Statement

This paper presents work whose goal is to advance the field of machine learning. There are many potential societal consequences of our work, none of which we feel must be specifically highlighted here.

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

# A. Appendix

## A.1. Computational Efficiency

Creat3r is data and computationally efficient. The efficiency stems from two aspects. The first one is active selection efficiency. Since the selection does not include 3DGS optimization, it takes less time to find the next best view. Average selection iteration costs of various methods are listed in Table A.1. Creat3r takes less than half the time needed in other counterparts. The duration is measured with a single NVIDIA V100 GPU. The second attribute is the optimization efficiency. Due to the robust geometry, Creat3r only uses half of the optimization time to converge.

*Table A.1.* Average selection iteration duration.

| Method | Time(sec) |
|---|---|
| FisherRF (Jiang et al., 2024) | 24.124 |
| Kopanas & Drettakis (2023) | 24.965 |
| Lyu et al. (2024) | 39.470 |
| Creat3r | 10.075 |

## A.2. More Quantitative Results

The complete evaluation of novel view synthesis is listed in Table A.2. For each baseline method, we evaluated with three different initialization strategies. For all the methods, initialization with Creat3r-MASt3R-matching consistently performs best among all the initializations, and Creat3r-LightGlue initialization performs better than COLMAP-induced subsampling initialization. We observe that MASt3R produces larger amounts of 2D correspondences than LightGlue, as it estimates more 3D points and is beneficial for active view selection. Creat3r [‡] and Creat3r [†] progressively estimate more and more points during the selection, leading to improvement by a large margin compared to other competitors.

## A.3. More Qualitative Results

Figure 5 presents more instances of novel view rendering with Creat3r and competing counterparts. When the selection is suboptimal, the 3D model cannot correctly render the novel views due to less exploration or a lack of finer detail. The renderings present artifacts or holes. Creat3r demonstrates renderings closest to the ground truth. Other methods have different kinds of artifacts.

## A.4. Selection Sequences Comparison

The view selection sequences generated by different approaches are visualized in Figure 6. The initial set, comprising the first three views, is adopted from the ReconFusion benchmark (Wu et al., 2024). The top row of Figure 6 il-

lustrates the selection process of Creat3r, which exhibits a spatially diverse distribution and progressively achieves comprehensive scene exploration. In contrast, the second and third rows—representing selections driven by the uncertainty estimates of Lyu *et al.* and FisherRF—reveal that while these methods explore the scene, they suffer from intermittent redundancy. Finally, the bottom row indicates that the approach of Kopanas & Drettakis results in a highly repetitive selection pattern.

## A.5. Visualization of Ablation Study

We visualize the qualitative impact of these design choices in Figure 7. Direct reliance on raw MASt3R predictions introduces geometric scale inconsistencies, yielding noisy and blurred renderings. As observed in Figure 7(b), relying solely on the confidence map biases the selection towards local detail refinement; this produces high-fidelity reconstruction in observed regions (e.g., the grass) but leaves the background largely unexplored and degraded. Conversely, utilizing only the exploration map ensures broader coverage of both foreground and background but fails to resolve high-frequency details, resulting in noticeable blurring. Finally, Creat3r synergizes both exploration and confidence objectives, achieving globally consistent and highly detailed reconstructions.

## A.6. Discussion of Hyperparameter Choices

The opacity parameter $o$ in Equation (9) serves as a constant rendering coefficient within the simplified Gaussian projection step. To evaluate the sensitivity of rendering quality to this parameter, we conduct an empirical ablation on the `garden` scene from the Mip-NeRF 360 dataset, varying $o$ across a wide operational range. As reported in Table A.3, the PSNR exhibits remarkable stability, yielding a standard deviation of merely 0.14 dB across all test configurations. This minimal variance demonstrates that the performance of Creat3r is highly robust to the initialization of the opacity coefficient. Consequently, we adopt a fixed value of $o = 0.8$ for all evaluations throughout this work.

## A.7. Generalization to Various Reconstruction Approaches

To demonstrate Creat3r's generalizability across different reconstruction methods, various evaluations are conducted. First, we integrate Creat3r with Depth Anything 3 (DA3) and conduct a comparative analysis using the `Bicycle` scene from the Mip-NeRF 360 dataset. We compare Creat3r-selected 10 views against 10 randomly sampled views. Notably, to address the known issue of DA3's difficulty with pre-estimated poses, we employ a transformation to align DA3's estimated poses with our selected viewpoints. The results of the novel view synthesis evaluation are shown in

*Table A.2.* Complete novel view synthesis evaluation on Mip-NeRF 360 dataset. The left and right columns show evaluations of 3DGS optimized with 10 and 20 selected views. (*) denotes methods initialized with COLMAP-induced subsampling points. ($\ddagger$) indicates methods initialized with Creat3r-LightGlue and ($\dagger$) indicates methods initialized with Creat3r-MASt3R. Best and second-best results are highlighted.

| Method | 10 cameras | | | 20 cameras | | |
|---|---|---|---|---|---|---|
| | PSNR ↑ | SSIM ↑ | LPIPS ↓ | PSNR ↑ | SSIM ↑ | LPIPS ↓ |
| FPS* | 12.529 | 0.261 | 0.613 | 14.918 | 0.389 | 0.528 |
| FPS$^\ddagger$ | 12.687 | 0.318 | 0.591 | 14.409 | 0.413 | 0.539 |
| FPS$^\dagger$ | 13.560 | 0.362 | 0.555 | 14.940 | 0.439 | 0.514 |
| FisherRF (Jiang et al., 2024)* | 12.625 | 0.264 | 0.608 | 15.434 | 0.390 | 0.515 |
| FisherRF (Jiang et al., 2024)$^\ddagger$ | 13.238 | 0.342 | 0.582 | 15.254 | 0.432 | 0.523 |
| FisherRF (Jiang et al., 2024)$^\dagger$ | 14.196 | 0.392 | 0.546 | 16.028 | 0.474 | 0.491 |
| Kopanas & Drettakis (2023)* | 13.022 | 0.284 | 0.596 | 15.658 | 0.407 | 0.506 |
| Kopanas & Drettakis (2023)$^\ddagger$ | 13.039 | 0.332 | 0.585 | 15.771 | 0.458 | 0.509 |
| Kopanas & Drettakis (2023)$^\dagger$ | 13.677 | 0.383 | 0.546 | 15.727 | 0.470 | 0.499 |
| Lyu et al. (2024)* | 12.561 | 0.264 | 0.612 | 15.446 | 0.401 | 0.518 |
| Lyu et al. (2024)$^\ddagger$ | 13.308 | 0.343 | 0.578 | 15.708 | 0.454 | 0.511 |
| Lyu et al. (2024)$^\dagger$ | 14.282 | 0.377 | 0.547 | 16.264 | 0.503 | 0.480 |
| Creat3r $^\ddagger$ | 16.040 | 0.449 | 0.536 | 19.637 | 0.567 | 0.443 |
| Creat3r $^\dagger$ | 17.809 | 0.511 | 0.523 | 20.678 | 0.601 | 0.397 |

*Table A.3.* Rendering quality across varying opacity configurations on the `garden` scene.

| Opacity | 0.1 | 0.5 | 0.8 | 1 |
|---|---|---|---|---|
| PSNR | 20.00 | 19.77 | 20.07 | 19.82 |

Table A.4, confirming that Creat3r consistently enhances performance even within different feed-forward frameworks. Second, we integrate Creat3r with 2DGS and evaluate the framework on the surface reconstruction task. The evaluation uses the `Truck` scene from Tanks&Temples dataset. The resulting F1-score is reported in Table A.5. While Creat3r selects 20 views for reconstruction, baseline methods such as 2DGS and 3DGS use the whole input set (more than 200 views). Even with a huge gap in the number of observed views, Creat3r consistently outperforms baseline models. Since 2DGS is designed to model geometry, Creat3r-2DGS surpasses Creat3r-3DGS with great improvement.

*Table A.5.* Surface Reconstruction with 3DGS and 2DGS. Creat3r only uses 20 views for optimization, while * indicates using all training views.

| | F1 |
|---|---|
| 3DGS* | 0.19 |
| Creat3r-3DGS | 0.19 |
| 2DGS* | 0.45 |
| Creat3r-2DGS | 0.47 |

*Table A.4.* Novel view synthesis with Depth Anything 3.

| Method | PSNR | SSIM | LPIPS |
|---|---|---|---|
| Random | 15.936 | 0.234 | 0.732 |
| Creat3r | 16.477 | 0.249 | 0.684 |

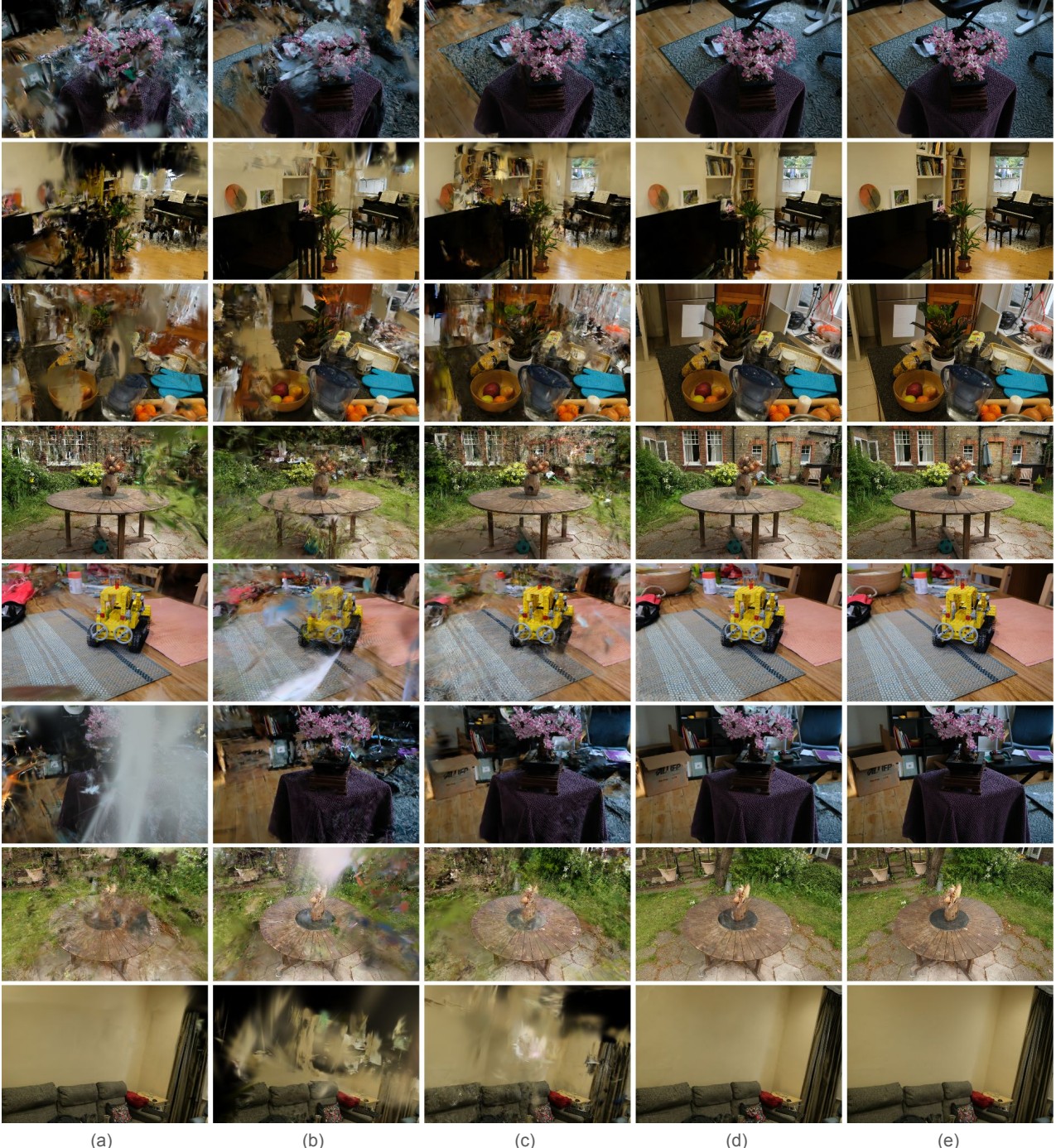

<table>
<tr><td>(a)</td><td>(b)</td><td>(c)</td><td>(d)</td><td>(e)</td></tr>
</table>

*Figure 5.* More qualitative comparisons of active view selection on the Mip-NeRF 360 dataset for 20 selected views. The results demonstrate novel view renderings produced by competing methods: (a) FisherRF (Jiang et al., 2024), (b) Kopanas & Drettakis (2023), (c) Lyu et al. (2024), and (d) our proposed method, Creat3r. Column (e) serves as the ground truth novel view reference. The comparison highlights the superior ability of Creat3r to synthesize high-fidelity views with finer details and fewer artifacts.

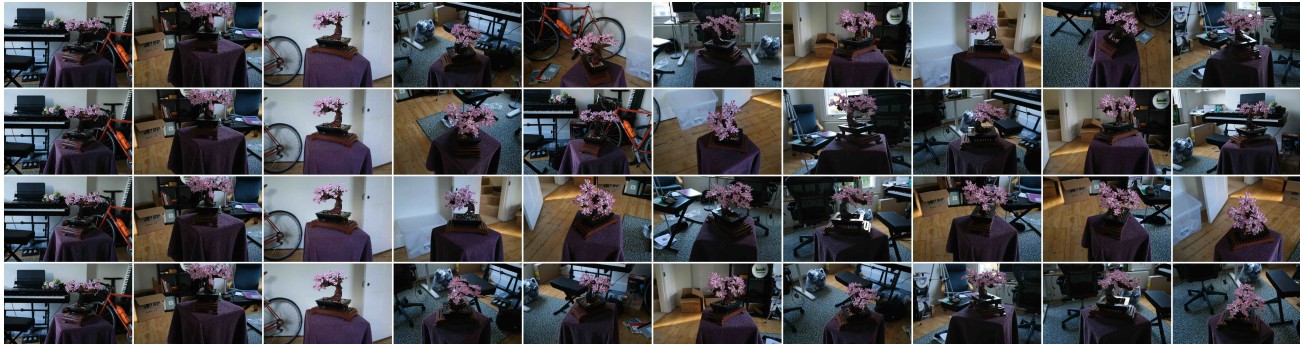

*Figure 6.* Visualization of active view selection sequences on the `bonsai` scene. The rows correspond to (from top to bottom): Creat3r, Lyu et al. (2024), FisherRF (Jiang et al., 2024), and Kopanas & Drettakis (2023). The first three columns display the fixed initial set, adopted from the ReconFusion benchmark (Wu et al., 2024).

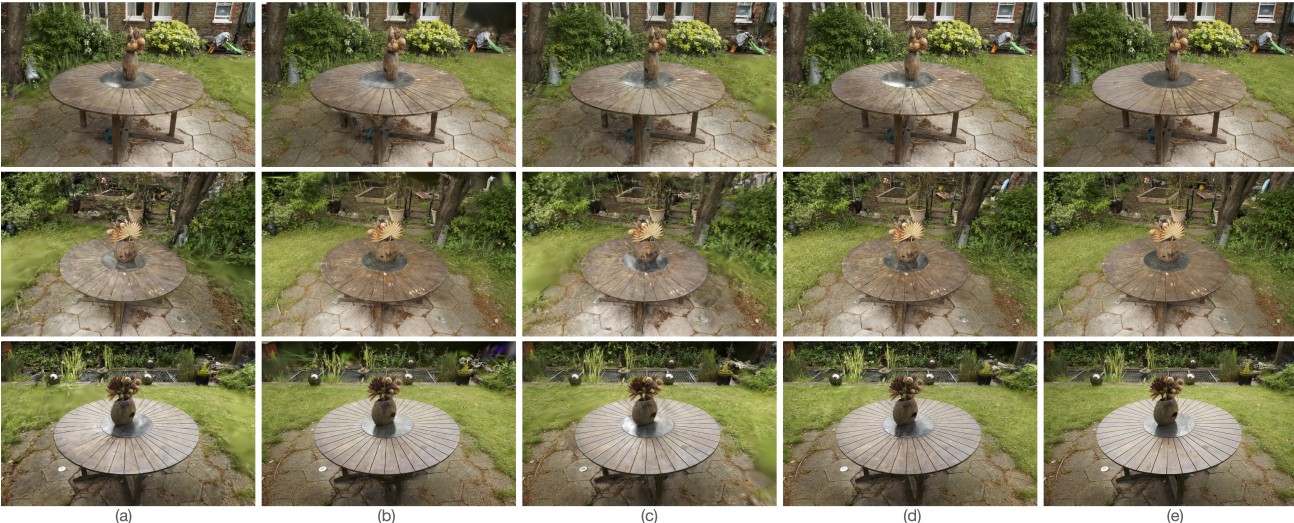

| (a) | (b) | (c) | (d) | (e) |

*Figure 7.* Qualitative evaluation of ablation components on the `garden` scene. The rendered novel views illustrate the impact of distinct design choices: (a) substituting robust point estimation with raw MASt3R predictions, (b) relying exclusively on confidence for selection, (c) relying exclusively on exploration for selection, and (d) the full Creat3r framework. The ground truth is provided in (e). This comparison highlights the specific contribution of each component and the superior reconstruction fidelity achieved by our holistic approach.

