# OpenReview forum: "Creat3r: Confidence Reaggregation for Exploration-aware Active 3D Reconstruction"
_ICML.cc/2026/Conference — ICML 2026 regular_

### Official Review · Reviewer_u2Tv · 2026-03-07

**Soundness:** 4
**Presentation:** 3
**Significance:** 3
**Originality:** 3
**Overall Recommendation:** 5
**Confidence:** 3

**Summary:**

This paper tackles the active 3D reconstruction. It first constructs an intermediate point cloud based on stereo geometry. Then it estimates the confidence map based on the photometric consistency and the exploration map based on the point density. Finally, the next best view is selected by maximizing the optimization goal which combines this two maps.

**Compliance With Llm Reviewing Policy:**

Affirmed.

**Final Justification:**

The author addressed my concern in hyperparameter. Overall, the work is solid and has great potential in real-world applications. I recommend acceptance.

**Key Questions For Authors:**

See weaknesses.

**Limitations:**

yes

**Strengths And Weaknesses:**

Strengths:
1. Active 3D reconstruction is an important task with 3D vision which enhances various downstream tasks.
2. The setting that new images are not accessible until a pose is selected is technically sound.
3. The method can be applied to any 3D reconstruction method, which broadens its real-world applications.
4. The method is simple and effective. The performance on Mip-Nerf 360 is SOTA.

Weaknesses:
1. The method can be applied to any 3D reconstruction methods, but only 3DGS is evaluated in this paper. Does the next best view policy work well with VGGT?
2. The choice of some hyper-parameters is not discussed in the paper, e.g. the opacity o in Eq 9, tau in Eq 10. Crucially, tau is an important hyperparameter, but its value is missing in the paper.

---

> ### Author Rebuttal · Authors · 2026-03-31
>
> We thank Reviewer u2Tv for the thoughtful comments.
>
> ---
> **W1:** *Does the next best view policy work well with VGGT?*
>
> We agree that the current submission only evaluates Creat3r with 3DGS, even though the paper positions the view-selection policy as reconstruction-backbone agnostic. In the current version, we chose 3DGS because it is the common reconstruction backbone used by the most relevant active-view-selection baselines, which enables direct and controlled comparison under the same protocol.
>
> More importantly, the proposed next-best-view policy is not tied to 3DGS-specific uncertainty estimation or optimization dynamics. Creat3r scores viewpoints using an intermediate geometry scaffold, a reaggregated 3D confidence field, and projected confidence and exploration maps, all of which are computed before downstream reconstruction and are conceptually independent of the final reconstruction module. This is also why the paper states that the framework is 3D-model agnostic and can accept other reconstruction techniques.
>
> Regarding VGGT specifically, VGGT is a feed-forward model that predicts scene geometry attributes such as camera parameters, depth maps, point maps, and tracks directly from multi-view images, rather than relying on iterative scene optimization. This makes it a plausible downstream reconstruction backend for our setting. Since Creat3r selects views based on coverage and reconstruction reliability of the currently observed scene geometry, rather than on 3DGS training behavior itself, we expect the policy to remain applicable when the selected views are passed to VGGT as input. That said, we agree that this claim is not yet validated experimentally in the paper. In the revision, we will clarify that backbone generality is a property of the selection policy design, while empirical validation beyond 3DGS remains an important direction for future work.
>
> ---
> **W2:** *Discussion on the hyperparameters.*
>
> We agree that the paper should discuss the hyperparameters more clearly, especially the trade-off parameter $\tau$ in Eq. (10) and the opacity $o$ in Eq. (9). In Creat3r, $o$ is used as a fixed rendering parameter in the simplified Gaussian projection step, while $\tau$ controls the balance between encouraging exploration of under-observed regions and penalizing already confident regions in the exploration score.
>
> The influence of $\tau$ has in fact been reflected in the ablation study in Section 4.5 and Table 4, although we agree that this connection should be stated more explicitly in the paper. In Table 4, we effectively examine three cases corresponding to $\tau=0$, $\tau=+\infty$, and $\tau=1$. When $\tau=+\infty$ (second row), the exploration term is neglected and the policy relies only on confidence, which yields better SSIM and LPIPS. When $\tau=0$ (third row), the confidence term is removed and the policy relies only on exploration, which gives higher PSNR. The full model uses $\tau=1$ (fourth row, highlighted), which provides the best overall balance by combining exploration and confidence. We agree, however, that the exact default value of $\tau$ should have been stated explicitly in the paper, and we will add this in the revision.
>
> For the opacity $o$, we additionally evaluated different values on the \texttt{garden} scene in Mip-NeRF 360:
>
> | Opacity | 0.1 | 0.5 | 0.8 | 1 |
> | :--- | :---: | :---: | :---: | :---: |
> | PSNR | 20.00 | 19.77 | 20.07 | 19.82 |
>
> The standard deviation across these settings is only $0.14$, indicating that the performance of Creat3r is robust to the choice of opacity. In our experiments, we use the fixed value $o=0.8$.
>
> In the revision, we will report the exact value of $\tau$ explicitly in the method section, clarify its relationship to the ablation results, and include a short discussion of opacity robustness. We appreciate the reviewer for identifying this presentation issue.

---

> > ### Author Rebuttal · Reviewer_u2Tv · 2026-04-01
> >
> > My concerns are well addressed. I keep my intial score.

---

> > > ### Author Response · Authors · 2026-04-02
> > >
> > > Reviewer u2Tv,
> > >
> > > We appreciate the final acknowledgement and the constructive feedback provided throughout the review process.
> > >
> > > The details regarding the hyperparameter values ($\tau$, $o$) and the robustness analysis discussed in the rebuttal will be integrated into the final version of the manuscript. We will also clarify the discussion on the reconstruction-backbone generality of our view-selection policy to ensure a more comprehensive presentation.
> > >
> > >
> > > Thank you,
> > >
> > > The Authors

---

### Official Review · Reviewer_aPFZ · 2026-03-10

**Soundness:** 3
**Presentation:** 3
**Significance:** 2
**Originality:** 2
**Overall Recommendation:** 5
**Confidence:** 3

**Summary:**

This paper presents **Creat3r**, an active NBV selection framework for 3DGS reconstruction. The core idea is to decouple viewpoint selection from iterative 3DGS optimization. Starting from a small seed set of image and pose pairs, Creat3r incrementally builds a sparse 3D scaffold via pairwise correspondences (LightGlue or MASt3R) and DLT triangulation. A 3D confidence field is constructed from multi-view photometric consistency and visibility, then projected to candidate viewpoints via a simplified Gaussian renderer to produce per-view *confidence and exploration maps*. These maps are combined into an *exploration measure* that balances refining uncertain regions and discovering unseen content. Experiments on Mip-NeRF 360 for NVS and Tanks & Temples for surface reconstruction show improvements over FisherRF, Lyu et al., and Kopanas & Drettakis, with substantially faster selection times.

**Compliance With Llm Reviewing Policy:**

Affirmed.

**Final Justification:**

The authors' rebuttal adequately addressed my concerns regarding the confidence field design, the exploration trade-off parameter, and correspondence dependence, including scaffold statistics and the integration of a more recent 3D foundation model (Depth Anything 3).

I therefore raise my score to 5 (Accept).

**Key Questions For Authors:**

**1. Confidence field alternatives.**

Have you tried incorporating triangulation angle or reprojection error into the confidence field? The current photometric consistency measure seems fragile for the Tanks & Temples scenarios highlighted.

**2. Sensitivity to $\tau$.**

How does performance vary with the trade-off parameter $\tau$? Does the optimal $\tau$ change across scenes or selection rounds? An ablation here would strengthen the paper.

**3. Scaffold statistics.**

How many 3D points does the scaffold contain at each selection round? How does this correlate with reconstruction quality? This would help readers understand the method's behaviour.

**Limitations:**

Yes

**Strengths And Weaknesses:**

**Strengths**

- **S1:** Identification and correction of a real protocol flaw.

The paper's most valuable conceptual contribution is pointing out the information leakage in existing active reconstruction benchmarks that initialize geometry via global SfM over the entire candidate pool. This is a genuine methodological issue, and the proposed leakage-free initialization is a welcome correction.

- **S2:** Clean, lightweight pipeline.

The overall framework is simple: correspondences &rarr; DLT triangulation &rarr; confidence field &rarr; Gaussian projection &rarr; exploration measure. The fact that Creat3r avoids re-optimizing 3DGS at each selection step is practically attractive and yields 2.5x faster selection iterations (Table A.1.). The pipeline is easy to understand and should be straightforward to reproduce.

- **S3:** Strong empirical gains.

The improvements over baselines are substantial (17.8 vs. 14.3 PSNR for the strongest competitor with 10 views). The surface reconstruction results on Tanks & Temples demonstrate that Creat3r's scaffold provides better coverage (25.9% recall vs. <10% for other baselines). The comparison with full 3DGS using more than 200 views is a nice feature showing that smart selection can match or outperform brute-force coverage.

**Weakness**

- **W1:** The confidence score is simplistic.

The confidence score (Eq. 8) is a product of average visibility and photometric consistency. This ignores several important factors such as triangulation angle (small baselines yield unreliable points even with high color consistency) or depth uncertainty for instance. The photometric consistency (Eq. 7) uses raw pixel color comparison, which is fragile under varying illumination, the condition that the authors highlight as challenging on Tanks & Temples (Section 4.4.). I would have expected a comparison with a more principled confidence measure such as reprojection error or triangulation angle for example.

- **W2:** The exploration measure (Eq. 10) is a simple linear combination with a fixed $\tau$.

The paper does not ablate the sensitivity to $\tau$, nor does it justify why a linear trade-off is appropriate. In practice, the optimal balance should be that early rounds explore more, and later rounds should refine. A single fixed $\tau$ seems suboptimal and the lack of any analysis is a gap.

- **W3:** Missing comparison with recent relevant work.

Several recent methods for active 3D reconstruction are not compared: GauSS-MI is cited in the related work but absent from experiments. The same for Ewen et al. (2025), which is only discussed textually. Given that both are recent concurrent works targeting the same problem, at least a runtime comparison or qualitative discussion of their experimental results would strengthen the paper.

- **W4:** Dependence on correspondence quality is under discussed.

The entire pipeline hinges on the quality of LightGlue / MASt3R correspondences. What happens when correspondences are sparse or unreliable, for example with textureless regions or repetitive structures? The paper shows LightGlue vs. MASt3R variants, but does not analyze failure modes. The gap between them is actually quite large (16.0 vs. 17.8 PSNR at 10 views), suggesting the method is sensitive to the matcher. This deserves more discussion, or more matchers could be tested.

- **W5:** Minor presentation issues.

The exploration measure in Eq. 10 uses a sum notation that is ambiguous: is it a sum over pixels? This should be clarified.

---

> ### Author Rebuttal · Authors · 2026-03-31
>
> We thank Reviewer aPFZ for the thoughtful comments.
>
> ---
>
> **W1&Q1:** *W1: Simple confidence score and Q1: Confidence field alternatives.*
>
>
> Our confidence reaggregation is more robust than a per-pair score because it is evaluated against the broader observed set $S^K$, not only the parent views of a triangulated point. In particular, a point produced under a small triangulation baseline or high depth uncertainty may still satisfy photometric consistency in its source pair, but it will often produce inconsistent projections across the wider observed set, driving $H(p)$ in Eqs. (7)--(8) toward zero. In this sense, the current confidence field already suppresses unreliable points through multi-view reaggregation. In addition, the exploration measure encourages diverse next-best-view selection, which progressively increases triangulation baselines and strengthens geometric constraints over later rounds. We agree that incorporating cues such as triangulation angle or reprojection error would be interesting extensions, and we will clarify this in the revision.
>
>
> **W2&Q2:** *W2: Simple exploration measure and Q2: Sensitivity to $\tau$.*
>
>
> The influence of $\tau$ is already reflected in the ablation study in Section 4.5 and Table 4, although we agree that this connection should be stated more explicitly. In Table 4, we effectively study three cases: $\tau=+\infty$, $\tau=0$, and $\tau=1$. When $\tau=+\infty$ (second row), the exploration term is removed, leading to better SSIM and LPIPS. When $\tau=0$ (third row), the confidence term is excluded, resulting in higher PSNR. The full model uses $\tau=1$ (fourth row, highlighted), which best synergizes exploration and confidence. We also note that a linear combination is a natural and transparent design here. Empirically, the exploration term dominates in the early rounds because many regions remain unobserved, whereas the confidence term becomes more influential in later rounds as the scaffold grows and refinement becomes more important. We will revise the paper to make the role and default value of $\tau$ explicit.
>
>
> **W3:** *Missing comparisons.*
>
> We thank the reviewer for this suggestion. GauSS-MI is conceptually related, but it assumes RGB-D inputs, whereas Creat3r estimates scene geometry from scratch using only RGB images. Ewen et al. (2025) is also relevant, but it was a concurrent work with no publicly available source code at the time of submission, making a direct quantitative comparison infeasible within our fully decoupled active-selection framework. We will clarify this positioning more explicitly in the revision.
>
>
> **W4&Q3:** *W4: Dependence on correspondence quality and Q3: Scaffold statistics.*
>
> We agree that correspondence quality is important for Creat3r. The scaffold statistics help explain the gap between Creat3r-MASt3R and Creat3r-LightGlue. In the 10-view Mip-NeRF 360 evaluation, MASt3R provides about 67k scaffold points on average by round 10, while LightGlue provides about 19k. This difference directly affects the quality of the scaffold, confidence field, and subsequent view selection. More generally, sparse matchers such as SuperGlue or LightGlue yield fewer correspondences, whereas denser approaches such as LoFTR, RoMa, or MASt3R can recover many more matches, including in weakly textured regions such as walls or ceilings. We will add this discussion and the scaffold statistics in the revision to better explain the observed performance gap and the dependence on correspondence density.
>
> **W5:** *Minor presentation issues.*
>
> We thank the reviewer for pointing this out. The summation in Eq. (10) is indeed taken over all pixels in the exploration map. We will improve the notation in the revised manuscript to ensure mathematical clarity.

---

> > ### Author Rebuttal · Reviewer_aPFZ · 2026-04-03
> >
> > Thank you for the strong rebuttal, it is much appreciated.
> >
> > My concerns have been fully resolved, hence I will increase my score.
> >
> > One remaining question is linked to the one of Reviewer wX3p regarding using MASt3R's confidence scores to select only confident 3D points, or even more recent backbones to build the 3D scaffold: do you have a threshold regarding the selected 3D points on their confidence? And the pipeline should be easily adaptable to other more recent 3D backbones such as Pi3 or Depth Anything 3, do you plan on adding new results as well using more recent and robust backbones?

---

> > > ### Author Response · Authors · 2026-04-03
> > >
> > > Reviewer aPFZ,
> > >
> > > We sincerely thank the reviewer for the positive feedback.
> > >
> > > **Regarding MASt3R Confidence Filtering:**
> > > While we explored using MASt3R’s confidence scores for point filtering, our empirical findings suggest that the primary bottleneck for 3D scaffold prediction is global geometric bias rather than point noise. Specifically, because MASt3R does not explicitly utilize known camera poses during inference, it often introduces inconsistent relative depth shifts between the foreground and background. This structural misalignment persists regardless of the confidence thresholding applied, making it a less stable geometric prior for view selection compared to our proposed approach.
> > >
> > > **Regarding More Recent 3D Backbones (Pi3 or Depth Anything 3):**
> > > We agree that the adaptability of our pipeline to evolving backbones is a key strength. As requested, we have integrated Creat3r with Depth Anything 3 (DA3). We conducted a comparative analysis on the *Bicycle* scene from the Mip-NeRF 360 dataset, comparing 10 views selected by Creat3r against 10 randomly sampled views. To address DA3's current issue with pre-estimated poses, we employed a $Sim(3)$ transformation to align DA3’s estimated poses with our selected viewpoints.  The results of the novel view synthesis evaluation are summarized below, confirming that Creat3r consistently enhances performance even when integrated with different feed-forward frameworks.
> > >
> > >
> > > | Method | PSNR ↑ | SSIM ↑ | LPIPS ↓ |
> > > | :--- | :--- | :--- | :--- |
> > > | **Creat3r** | **16.477** | **0.249** | **0.684** |
> > > | Random | 15.936 | 0.234 | 0.732 |
> > >
> > > The new evaluation results using more recent and robust backbones will be carefully integrated into the final version of the manuscript.
> > >
> > > Thank you,
> > >
> > > The Authors

---

### Official Review · Reviewer_wX3p · 2026-03-12

**Soundness:** 2
**Presentation:** 2
**Significance:** 2
**Originality:** 3
**Overall Recommendation:** 4
**Confidence:** 4

**Summary:**

The authors address the problem of active view selection. The goal is to identify a small subset of informative camera views that reduces the number of acquisitions while enabling an accurate 3D reconstruction. Starting from a predefined set of observed camera poses, the authors propose to build a 3D confidence field based on photometric information that is rendered to candidate poses to generate 2D confidence and exploration maps. By maximizing a weighted difference between exploration and confidence maps, the authors identify the next best camera view for their set of observed cameras. Subsequently, this set can be used for 3D reconstruction, e.g., with 3DGS. The authors evaluate their strategy, Creat3r, on two datasets for two tasks: MipNeRF-360 for 3D reconstruction and Tanks&Temples for surface reconstruction.

**Compliance With Llm Reviewing Policy:**

Affirmed.

**Final Justification:**

The strong rebuttal addressed my concerns, so I raised the score accordingly.

**Key Questions For Authors:**

- How does Creat3r perform against more recent baselines such as POp-GS [A] and Gauss-MI [B]?
- Can Creat3r’s NBV also benefit feed-forward reconstruction models such as VGGT [C]?
- How does Creat3r handle camera views that gather many projected points that would normally not be observed from that particular view due to occlusions (e.g., when 1 side of an object is not well-observed while others are, the camera view pointing towards the unobserved object side still gathers many points)?
- How would Creat3r’s reconstruction quality be affected when using a 3DGS optimized Gaussian field (maybe one suitable for utilizing sparse views) for rendering instead of the simplistic projection?

(References [A-C] are in the section above.)

**Limitations:**

The authors discussed the limitations of their work well. Still, it would be beneficial to also highlight two additional aspects:
- Creat3r is quite dependent on the off-the-shelf matching framework it uses.
- Even though the authors claim that their method works well for varying exposure, their NBV strategy is only based on the total number of projected points and how well the photometric measurements from a point observed from multiple views agree with each other; therefore, the model should be prone to illumination changes and view-dependent effects.

**Strengths And Weaknesses:**

Strengths:
- The paper’s introduction and related work are clearly written and easy to follow, and the problem setting is well motivated (presentation).
- The authors propose a simple, novel approach to address the problem of active view selection (originality). Compared to prior methods, Creat3r enables NBV without relying on a specific reconstruction strategy, thereby achieving general applicability and faster runtime (significance). Additionally, Creat3r outperforms all baselines reported in the paper.

Weaknesses:
- **W1: Missing baselines (soundness).** The authors should also compare against more recent baselines such as POp-GS [A] and Gauss-MI [B].
- **W2: Missing ablations (soundness).** The authors should ablate the selection of their fixed opacity value o and their weighting parameter \theta.
- **W3: Overclaim performance (soundness).** The authors claim that their method is capable of reconstructing the surface under limited views. Even though Creat3r seems better than the baselines, its low performance in Precision, Recall, and F1 does not justify its ability to perform surface reconstruction.
- **W4: Comparisons could improve (soundness).** For a more complete/fair comparison, the authors should also report the original method of Kopanas & Drettakis, based on NeRF, rather than only using a self-implemented variant based on 3DGS.
- **W5: Model agnostic claim (soundness).** The authors claim that their method is model-agnostic. While it is true that they do not depend on the 3D model/optimization paradigm in theory, it would be interesting to see whether this method would also benefit NeRFs or feed-forward reconstruction models, e.g., VGGT [C].
- **W6: Suboptimal presentation/clarity (presentation).**
    - a) The related work could be explicitly divided into subsections covering, e.g., active view selection for 3D reconstruction, information-theoretical view quantification.
    - b) Section 3.1 can be largely shortened, as the presented knowledge is a standard baseline in multi-view geometry, in favor of additional experiments or limitations that are currently included in the supplementary.
    - c) Some parts in the method section lack clarity:
The authors should clarify what they mean by projected footprint;
If r_2D is chosen such that every Gaussian occupies one pixel, how is equation 9 different from simple point projection?
Formulations such as “projected footprint “or “unit intensity” should be clarified. For the latter, e.g., one could define a binary function which is 1 if a point projects to pixel location p and 0 otherwise.
Some notation is not introduced, e.g., \bar{M}_Conf(W); they probably mean the average, but clarification would be helpful.
   - d) The authors should clarify the statement: “This suggests that uncertainty estimation, either with information gain or variational inference, is more beneficial from robust geometry, while Kopanas & Drettakis is less dependent on geometry.”
   - e) For table 4, the authors should specify how they used the predicted MASt3R point cloud (which confidence value for thresholding? Triangulation of matches or unprojected depth map)
- **W7: Unclear method figure (presentation).** The method figure would benefit from reiteration, as it does not really help the reader understand the presented method. (Minor: the red camera is not aligned with the selected NBV image content)
- **W8: Surface reconstruction missing (soundness).** Qualitative results showing Creat3r’s surface reconstruction would be interesting.
- **W9: Limited impact with highly specific setting (soundness).** Although the proposed strategy is interesting, the impact appears limited because the addressed domain is so highly specialized.

Minor:
- The authors miss the dataset citations for Mip-NeRF and Tank&Temples.

References:
- [A] Wilson J, Almeida M, Mahajan S, Labrie M, Ghaffari M, Ghasemalizadeh O, Sun M, Kuo CH, Sen A. Pop-gs: Next best view in 3d-gaussian splatting with p-optimality. In CVPR 2025.
- [B] Xie Y, Cai Y, Zhang Y, Yang L, Pan J. Gauss-mi: Gaussian splatting Shannon mutual information for active 3d reconstruction. In RSS 2025.
- [C] Wang J, Chen M, Karaev N, Vedaldi A, Rupprecht C, Novotny D. Vggt: Visual geometry grounded transformer. In CVPR 2025.

---

> ### Author Rebuttal · Authors · 2026-03-31
>
> We thank Reviewer wX3p for the thoughtful comments.
>
> **W1&Q1:** GauSS-MI is conceptually related, but it assumes RGB-D inputs, whereas Creat3r estimates geometry from scratch using only RGB images. POp-GS is also relevant, but it was concurrent work without publicly available code at the time of submission, which made a direct quantitative comparison infeasible within our fully decoupled active-selection pipeline. We will clarify this positioning more explicitly in the revision.
>
> **W2:** We agree that the hyperparameter discussion should be clearer. The effect of $\tau$ in Eq.(10) is already reflected in Table4: the confidence-only case corresponds to $\tau=+\infty$, the exploration-only case to $\tau=0$, and the full model uses $\tau=1$. Empirically, $\tau=+\infty$ gives better SSIM/LPIPS, $\tau=0$ gives higher PSNR, and $\tau=1$ provides the best balance. For the opacity $o$, we additionally evaluated $o\in\{0.1,0.5,0.8,1.0\}$ on the *Garden* scene in Mip-NeRF 360, obtaining PSNR values $20.00, 19.77, 20.07,$ and $19.82$, respectively. The standard deviation is only $0.14$, indicating that Creat3r is robust to the choice of opacity. We will report these details explicitly in the revision.
>
> **W3:** In the paper, we follow prior work and use 3DGS as the common reconstruction backbone for controlled comparison. However, Creat3r is not restricted to vanilla 3DGS and can also be paired with geometry-oriented variants such as 2DGS. For example, on the *Truck* scene in Tanks&Temples, standard 2DGS optimized with 251 images achieves an F1 score of $0.45$, while Creat3r-2DGS optimized with only 20 selected images achieves $0.47$. We will revise the wording to avoid overstating the claim and make the scope clearer.
>
> **W4:** For fairness, we follow the benchmark and unified 3DGS-based evaluation codebase released by Lyu et al., which is also used by prior active reconstruction methods. We will clarify this protocol choice more explicitly in the paper.
>
> **W5&Q2:** Creat3r decouples view selection from 3D model optimization, which makes the policy naturally compatible with other reconstruction backbones. In particular, feed-forward methods such as VGGT could use Creat3r as an active data collection stage before inference. This could be beneficial because the policy selects a compact but informative set of views without relying on iterative scene optimization. At the same time, current NVS and surface-reconstruction benchmarks typically assume pre-estimated camera poses, while VGGT is image-only; a direct comparison under the same protocol would therefore be unfair. We will clarify this distinction in the revision.
>
> **W6:** We thank the reviewer for the constructive suggestions. In the revision, we will reorganize the presentation and improve the mathematical definitions. In particular, the effective region of each point is set so that its footprint is approximately one pixel in its parent views, while its projected 2D radius in other views depends on depth; therefore a simple binary visibility function is insufficient to describe the rendered maps. We will also clarify that robust initialization benefits uncertainty-based methods as well: FisherRF and Lyu et al. improve noticeably when initialized with Creat3r points. Finally, in the ablation, the alternative scaffold is formed by directly using MASt3R-predicted point maps.
>
> **W7&W8:** We agree that both the method figure and the qualitative comparisons can be improved. We will refine these visualizations and strengthen the presentation of the surface-reconstruction results in the revised manuscript.
>
> **W9:** We respectfully believe the applicability is broader than the benchmark setting. In practice, Creat3r can serve as a pre-processing or automatic data collection stage before downstream NeRF/3DGS optimization or feed-forward geometry inference. Such a process can be integrated with robots or drones to reduce manual data capture and improve acquisition efficiency.
>
> **Q3:** Yes. In this case, a new view may still project many existing points even when observing the object from a less-covered side, which is common in early rounds when the known set is small. After that image is acquired, new points are triangulated from it and added to the scaffold. Meanwhile, because the newly visible side is weakly supported by the current scaffold, the projected confidence remains low there, which helps the exploration measure prioritize such views.
>
> **Q4:** Creat3r cannot be replaced directly by sparse-view 3DGS optimization for two reasons. First, sparse-view 3DGS still depends heavily on appearance consistency, and can fail in challenging scenes where exposure varies substantially across views. Second, coupling optimization with view selection reduces efficiency and reintroduces the same computational burden as prior optimization-based baselines. Our simplified projection is therefore a deliberate design choice for efficient, optimization-decoupled selection.

---

> > ### Author Rebuttal · Reviewer_wX3p · 2026-04-02
> >
> > Thank you for your comments. Please find my responses below.
> >
> > **W1&Q1**: GauSS-MI uses RGB-D, while the final Creat3r is RGB only (even though there is one row in the author's ablation that uses point maps from MASt3R); so it is ok not to compare to it. On the other hand, POp-GS is not really concurrent work given that it was published at CVPR 2025. But indeed, without publicly available code, it is more challenging. Still, the authors could have compared with POp-GS in POp-GS's setting.
> >
> > **W2**: Addressed
> >
> > **W3**: Addressed
> >
> > **W4**: Addressed
> >
> > **W5&Q2**: The authors argue that it is not easy to experimentally verify that Creat3r benefits feed-forward 3D reconstruction models such as VGGT. I agree that VGGT's output cannot be directly evaluated on rendering quality, as it predicts camera pose, depth, point maps, and tracks. However, by evaluating Creat3r+2DGS, they experimentally show that Creat3r is not restricted to vanilla 3DGS and provides a benefit for surface reconstruction.
> > Alternatively, one could have used a feed-forward 3D reconstruction model that can also predict Gaussians, such as Depth Anything 3, which would make it more straightforward to evaluate Creat3r's impact on feed-forward reconstruction models.
> >
> > **W6**: Addressed with follow-up question on e): As I understood, one of the authors' baselines takes the raw MASt3R-predicted point cloud as a scaffold. Would this variant improve if one thresholds the MASt3R-predicted point cloud by the MASt3R-predicted confidence to extract more robust points?
> >
> > **W7&W8**: Will be addressed in the paper, ok.
> >
> > **W9**: Addressed
> >
> > **Q3**: Addressed
> >
> > **Q4**: Addressed
> >
> > I plan to update my score accordingly.

---

> > > ### Author Response · Authors · 2026-04-03
> > >
> > > Reviewer wX3p,
> > >
> > > We sincerely thank the reviewer for the final acknowledgment and the constructive feedback.
> > >
> > > W1&Q1 (POp-GS): We thank the reviewer for this helpful clarification. We agree that POp-GS is not concurrent work, and our earlier wording was imprecise. Our main concern is instead the comparability of the evaluation protocol. Based on the FisherRF results reported in the POp-GS paper, it appears possible that their Mip-NeRF 360 evaluation follows the standard protocol with global-SfM initialization. Since our work specifically revisits this protocol and argues that such initialization may introduce information leakage, we were cautious about making a direct comparison without being able to verify that both methods were evaluated under matched conditions. In our view, the most informative comparison would be to test POp-GS under the same leakage-free initialization used in our paper. As the implementation was not publicly available at the time of submission, we were unfortunately unable to carry out that controlled comparison. We appreciate the reviewer’s point, and we would be very interested in reevaluating POp-GS under our initialization protocol once code becomes available.
> > >
> > > W5&Q2 (Feed-forward models): We appreciate the suggestion. To demonstrate Creat3r's generalizability across different feed-forward architectures, we have integrated it with Depth Anything 3 (DA3) and conducted a comparative analysis using the *Bicycle* scene from the Mip-NeRF 360 dataset. We compared Creat3r-selected 10 views against 10 randomly sampled views. Notably, to address the known issue of DA3's difficulty with pre-estimated poses, we employ a $ Sim(3) $ transformation to align DA3’s estimated poses with our selected viewpoints. The results of the novel view synthesis evaluation are summarized below, confirming that Creat3r consistently enhances performance even within different feed-forward frameworks.
> > >
> > > | Method | PSNR ↑ | SSIM ↑ | LPIPS ↓ |
> > > | :--- | :---: | :---: | :---: |
> > > | **Creat3r** | **16.477** | **0.249** | **0.684** |
> > > | Random | 15.936 | 0.234 | 0.732 |
> > >
> > > W6 (MASt3R Filtering): Regarding the confidence thresholding, we empirically found that the primary bottleneck for the MASt3R-predicted scaffold is not just point noise, but global geometric bias. Specifically, MASt3R often introduces relative depth shifts between foreground and background because it does not explicitly utilize known poses during inference. This bias makes it a less stable scaffold compared to our approach, regardless of confidence thresholding.
> > >
> > > All discussed points and clarifications will be carefully integrated into the final version of the paper.
> > >
> > > Thank you,
> > >
> > > The Authors

---

### Official Review · Reviewer_ce2L · 2026-03-13

**Soundness:** 4
**Presentation:** 4
**Significance:** 2
**Originality:** 4
**Overall Recommendation:** 5
**Confidence:** 4

**Summary:**

This paper proposes a novel method for active viewpoint selection when training 3DGS. Unlike previous methods, which relied on a coupled iterative framework where GS optimization was done every time a new viewpoint was selected, this paper introduces learnable confidence and exploration maps that are used to reliably select the next best view. Experiment results show faster convergence with better performance on novel view synthesis against previous methods.

[1] “3D Gaussian Splatting for Real-Time Radiance Field Rendering”, SIGGRAPH 2023.

**Compliance With Llm Reviewing Policy:**

Affirmed.

**Final Justification:**

The initial manuscript is already strong enough for acceptance. I therefore keep my score (5).

**Key Questions For Authors:**

- As mentioned in the weaknesses section, I have my doubts regarding the significance of selecting the next best view during 3DGS optimization. I am curious to hear the authors’ perspectives on this.
- Selecting the next best informative viewpoint during data collection, on the other hand, is of much more significance, in my opinion. To enable this, we would require estimating the exact pose of the next viewpoint instead of selecting from a subset of pre-defined poses. Is there a way you could obtain such a pose (or an alternative strategy)? Since such an experiment cannot be performed on real scenes, any experiments on simulation datasets should suffice.

**Limitations:**

yes

**Strengths And Weaknesses:**

**Strengths:**

- To the best of my knowledge, the idea of using an exploration-exploitation map to select the next best view is completely novel in this area.
- The proposed framework bypasses the need to optimize Gaussians every time a new viewpoint is selected, which is a huge improvement over previous methods.
- Experiment results validate the effectiveness of the proposed framework.
- The paper is well presented and easy to follow through.

**Weaknesses:**

- While the idea of active viewpoint selection to decrease data acquisition burdens seems ideal, it is non-trivial how it could be implemented during the data collection process. Pioneering works focused on NeRF-like frameworks [1], which took hours to optimize, and thus the necessity of selecting the best views during training was more significant. Recent Gaussian-based frameworks have achieved significant reductions in optimization runtime [2]. Thus, the significance of this task in terms of optimization speed seems to have decreased significantly.

[1] “ActiveNeRF: Learning where to See with Uncertainty Estimation”, ECCV 2022.\
[2] “FastGS: Training 3D Gaussian Splatting in 100 Seconds”, CVPR 2026.

---

> ### Author Rebuttal · Authors · 2026-03-31
>
> We thank Reviewer ce2L for the thoughtful comments.
>
> ---
>
> **Q1:**  *”… significance of selecting the next best view during 3DGS optimization”*
>
> Our primary motivation is data efficiency under a limited acquisition budget: given that only a small number of views can be acquired, which views should be selected to maximize final reconstruction quality? This question remains important even if the downstream reconstruction engine becomes substantially faster. In many realistic settings, the bottleneck is not only optimization, but also image acquisition, robot motion, storage, bandwidth, and the practical difficulty of capturing dense multi-view coverage. From this perspective, next-best-view selection is valuable because it reduces the number of required observations while preserving or improving reconstruction quality.
>
> Our experiments support this interpretation. On Mip-NeRF 360, Creat3r is evaluated in sparse-view regimes with only 10 or 20 selected views, and consistently outperforms prior selection strategies in PSNR, SSIM, and LPIPS. On Tanks\&Temples, it also achieves the best F1 score for surface reconstruction under the same limited-view setting. These results suggest that the benefit of active view selection is not merely faster training, but better reconstruction quality with fewer acquired views.
>
> We also emphasize that Creat3r is intentionally decoupled from iterative 3DGS optimization during selection. This means our method does not depend on repeated expensive optimization loops to estimate uncertainty or score candidate views. As a result, the method remains relevant even as reconstruction backbones become faster, since the core problem we address is not tied to one specific training runtime, but to selecting informative observations efficiently. In the revision, we will clarify that our contribution should be understood primarily as an approach to acquisition-efficient sparse-view reconstruction, with computational savings as a secondary benefit rather than the sole motivation.
>
> ---
>
> **Q2:**  *”Selecting the next best informative viewpoint during data collection …”*
>
>
> We agree that selecting the next informative viewpoint during data collection is a highly meaningful setting, and that moving beyond a fixed candidate pool toward estimating a new camera pose is an important next step.
>
> In the current paper, we focus on the standard protocol where a pool of candidate poses is given in advance, and the algorithm selects the next view from that pool. We chose this setting for two reasons. First, it enables controlled comparison with prior active reconstruction methods. Second, it lets us isolate the selection problem while explicitly avoiding information leakage from unseen images or global SfM computed from the full candidate set. Thus, our present contribution is a leakage-free and optimization-decoupled selection strategy under the discrete-pool setting.
>
> Meanwhile, our framework can be extended naturally to online viewpoint generation. Creat3r maintains an intermediate geometry scaffold and a 3D confidence field, and evaluates viewpoints through projected confidence and exploration maps. This same idea could be applied beyond a fixed pool by scoring poses sampled from a continuous camera space. A practical strategy is a two-stage scheme: first generate coarse candidate poses from a simulator, motion planner, or local camera search around under-observed regions; then refine the best pose by locally perturbing position and orientation and re-evaluating the same exploration score. Since our criterion depends on the current scaffold rather than repeated full 3DGS re-optimization, it is well-suited to this type of online pose search.
>
> We agree that experiments on simulation datasets would be an appropriate way to study this extension. Although such experiments are beyond the scope of the current submission, we will add this discussion in the revision and clarify that extending Creat3r from discrete-pool selection to continuous pose generation is a promising future direction for embodied or robotic data collection.

---

> > ### Author Rebuttal · Reviewer_ce2L · 2026-04-04
> >
> > All of my concerns have been well addressed.

---

> > > ### Author Response · Authors · 2026-04-04
> > >
> > > Reviewer ce2L,
> > >
> > > We appreciate the final acknowledgement and the constructive feedback.
> > >
> > > In the final version of the manuscript, we will carefully emphasize that Creat3r's primary significance lies in acquisition-efficient sparse-view reconstruction. We will also incorporate the discussion on extending our framework from discrete-pool selection to continuous pose generation, highlighting its potential for embodied environments.
> > >
> > > Thank you,
> > >
> > > The Authors

---

### Decision · Program_Chairs · 2026-04-30

**Decision:**

Accept (regular)

**Comment:**

This submission eventually got four positive recommendations. Initially, the reviewers were concerned about the evaluation, the presentation, and runtime efficiency. The authors did a good job and addressed most of these concerns in the rebuttal. During the discussion among the authors and the reviewers, the reviewers confirmed that their concerns had been fully addressed. Thus, all reviewers reached a consensus without a discussion. The AC read through the manuscript, all reviews, the rebuttal, and the discussions among the authors and the reviewers, the AC agreed with all reviewers, and liked the idea of the paper. Per these, the AC made a decision of acceptance. This decision was approved by the SAC as well.